# Uncoupled Regression
# from Pairwise Comparison Data

**Liyuan Xu** [1,2] *
liyuan@ms.k.u-tokyo.ac.jp
**Gang Niu** [2]
gang.niu@riken.jp

**Junya Honda** [1,2]
honda@stat.t.u-tokyo.ac.jp
**Masashi Sugiyama** [2,1]
sugi@k.u-tokyo.ac.jp

[1]The University of Tokyo     [2]RIKEN

## Abstract

*Uncoupled regression* is the problem to learn a model from unlabeled data and the set of target values while the correspondence between them is unknown. Such a situation arises in predicting anonymized targets that involve sensitive information, e.g., one's annual income. Since existing methods for uncoupled regression often require strong assumptions on the true target function, and thus, their range of applications is limited, we introduce a novel framework that does not require such assumptions in this paper. Our key idea is to utilize *pairwise comparison data*, which consists of pairs of unlabeled data that we know which one has a larger target value. Such pairwise comparison data is easy to collect, as typically discussed in the learning-to-rank scenario, and does not break the anonymity of data. We propose two practical methods for uncoupled regression from pairwise comparison data and show that the learned regression model converges to the optimal model with the optimal parametric convergence rate when the target variable distributes uniformly. Moreover, we empirically show that for linear models the proposed methods are comparable to ordinary supervised regression with labeled data.

## 1 Introduction

In supervised regression, we need a vast amount of labeled data in the training phase, which is costly and laborious to collect in many real-world applications. To deal with this problem, weakly-supervised regression has been proposed in various settings, such as semi-supervised learning (see Kostopoulos et al. [17] for the survey), multiple instance regression [27, 34], and transductive regression [4, 5]. See [35] for a thorough review of the weakly-supervised learning in binary classification, which can be extended to regression with slight modifications.

Uncoupled regression [2] is one variant of weakly-supervised learning. In ordinary "coupled" regression, the pairs of features and targets are provided, and we aim to learn a model that minimizes a certain prediction error on test data. On the other hand, in the uncoupled regression problem, we only have access to unlabeled data and the set of target values, and thus, we do not know the true target for each data point. Such a situation often arises when we aim to predict people's sensitive matters such as one's annual salary or the total amount of deposit, the data of which is often anonymized for privacy concerns. Note that it may not be impossible to conduct uncoupled regression without further assumptions, since no labeled data is provided.

Carpentier and Schlueter [2] showed that uncoupled regression is solvable when the feature is one-dimensional and the true target function is monotonic to it. Although their algorithm is of less

practical use due to its strong assumption, their work offers a valuable insight that a model is learnable from uncoupled data if we know the ranking in the dataset. In this paper, we show that, instead of imposing the monotonic assumption, we can infer such ranking information from data to solve uncoupled regression. We use *pairwise comparison data* as a source of ranking information, which consists of the pairs of unlabeled data that we know which data point has a larger target value.

Note that pairwise comparison data is easy to collect even for sensitive matters such as one's annual earnings. Although people often hesitate to give explicit answers to it, it might be easier to answer indirect questions: "Which person earns more than you?" [2], which yields pairwise comparison data that we need. The difficulty here is that comparison is based on the target value, which contains the noise. Hence, the comparison data is also affected by this noise. Considering that we do not put any assumption on the true target function, our methods are applicable to many situations. A similar problem was considered in Bao et al. [1] as well.

One naive method for uncoupled regression with pairwise comparison data is to use a score-based ranking method [29], which learns a score function with the minimum inversions in pairwise comparison data. With such a score function, we can match unlabeled data and the set of target values, and then, conduct supervised learning. However, as discussed in Rigollet and Weed [28], we cannot consistently recover the true target function even if we know the true order of missing target values in unlabeled data due to the noise in them.

In contrast, our methods directly minimize the regression risk. We first rewrite the regression risk so that it can be estimated from unlabeled and pairwise comparison data, and learn a model through empirical risk minimization. Such an approach based on risk rewriting has been extensively studied in the classification scenario [7, 6, 23, 30, 18] and exhibits promising performance. We propose two estimators of the risk defined based on the expected Bregman divergence [11], which is a natural choice of the risk function. We show that if the marginal distribution of the target variable is uniform then the estimators are unbiased and the learned model converges to the optimal model with the optimal rate. In general cases, however, we prove that it is impossible to have such an unbiased estimator in any marginal distributions and the learned model may not converge to the optimal one. Still, our empirical evaluations based on synthetic data and benchmark datasets show that our methods exhibit similar performance to a model learned from coupled data for ordinary supervised regression.

The paper is structured as follows. After discussing the related work in Section 2, we formulate the uncoupled regression problem with pairwise comparison data in detail in Section 3. In Sections 4 and 5, we discuss two methods for uncoupled regression and derive estimation error bounds for each method. Finally, we show empirical results in Section 6 and conclude the paper in Section 7.

## 2   Related work

Several methods have been proposed to match two independently collected data sources. In the context of data integration [3], the matching is conducted based on some contextual data provided for both data sources. For example, Walter and Fritsch [31] used spatial information as contextual data to integrate two data sources. Some work evaluated the quality of matching by some information criterion and found the best matching by maximizing the metric. This problem is called cross-domain object matching (CDOM), which was formulated in Jebara [15]. A number of methods have been proposed for CDOM, such as Quadrianto et al. [26], Yamada and Sugiyama [33], and Jitta and Klami [16].

Another line of related work in the uncoupled regression problem imposed an assumption on the true target function. For example, Carpentier and Schlueter [2] assumed that the true target function is monotonic to a single feature, and it was refined theoretically by Rigollet and Weed [28]. Another common assumption is that the true target function is exactly expressed as a linear function of the features, which was studied in Hsu et al. [14] and Pananjady et al. [24]. Although the model learned from these methods converges to the true target function with infinite uncoupled data, they are of less practical use due to their strong assumptions. On the other hand, our methods do not require any assumptions on such mapping functions and are applicable to wider scenarios.

It is worth noting that some methods use uncoupled data to enhance the performance of semi-supervised learning. For example, in label regularization [19], uncoupled data is used to regularize a regression model so that the distribution of prediction on unlabeled data is close to the marginal distribution of target variables, which was reported to increase the accuracy.

Pairwise comparison data was originally considered in the ranking problem [29, 22], which aims to learn a score function that can rank data correctly. In fact, we can apply ranking methods, such as rankSVM [13], to our problem. However, the naive application of them performs inferiorly compared to proposed methods, as we will show empirically, since our goal is not to order data correctly but to predict true target values.

## 3    Problem settings

In this section, we formulate the uncoupled regression problem and introduce pairwise comparison data.

### 3.1    Uncoupled regression problem

We first formulate the standard regression problem briefly. Let $\mathcal{X} \subset \mathbb{R}^d$ be a $d$-dimensional feature space and $\mathcal{Y} \subset \mathbb{R}$ be a target space. We denote $\boldsymbol{X}, Y$ as random variables on spaces $\mathcal{X}, \mathcal{Y}$, respectively. We assume these random variables follow the joint distribution $P_{\boldsymbol{X},Y}$. The goal of the regression problem is to obtain model $h : \mathcal{X} \to \mathcal{Y}$ in hypothesis space $\mathcal{H}$ which minimizes the risk defined as

$$R(h) = \mathbb{E}_{\boldsymbol{X},Y}\left[l(h(\boldsymbol{X}), Y)\right], \tag{1}$$

where $\mathbb{E}_{\boldsymbol{X},Y}$ denotes the expectation over $P_{\boldsymbol{X},Y}$ and $l : \mathcal{Y} \times \mathcal{Y} \to \mathbb{R}_+$ is a loss function.

The loss function $l(z, t)$ measures the closeness between a true target $t \in \mathcal{Y}$ and an output of a model $z \in \mathcal{Y}$, which generally grows as prediction $z$ gets far from the target $t$. In this paper, we mainly consider $l(z, t)$ to be the Bregman divergence $d_\phi(t, z)$, which is defined as

$$d_\phi(t, z) = \phi(t) - \phi(z) - (t - z)\phi'(z)$$

for some convex function $\phi : \mathbb{R} \to \mathbb{R}$, and $\phi'$ denotes the derivative of $\phi$. It is natural to have such a loss function since the minimizer of risk $R$ is $\mathbb{E}_{Y|\boldsymbol{X}=\boldsymbol{x}}[Y]$ when hypothesis space $\mathcal{H}$ is rich enough [11], where $\mathbb{E}_{Y|\boldsymbol{X}=\boldsymbol{x}}$ is the conditional expectation over the distribution of $Y$ given $\boldsymbol{X} = \boldsymbol{x}$. Many common loss functions can be interpreted as the Bregman divergence; for instance, when $\phi(x) = x^2$, then $d_\phi(t, z)$ becomes the $l_2$-loss, and when $\phi(x) = x \log x - (1 - x)\log(1 - x)$, then $d_\phi(t, z)$ is the Kullback–Leibler divergence between the Bernoulli distributions with probabilities $t$ and $z$.

In the standard regression scenario, we are given labeled training data $\mathcal{D} = \{(\boldsymbol{x}_i, y_i)\}_{i=1}^n$ drawn independently and identically from $P_{\boldsymbol{X},Y}$. Then, based on the training data, we empirically estimate risk $R(h)$ and obtain model $\hat{h}$ by the minimizing the empirical risk. On the other hand, in uncoupled regression, what we are given are unlabeled data $\mathcal{D}_{\mathrm{U}} = \{\boldsymbol{x}_i\}_{i=1}^{n_{\mathrm{U}}}$ and target values $\mathcal{D}_Y = \{y_i\}_{i=1}^{n_Y}$ without correspondance. Here, $n_{\mathrm{U}}$ is the size of unlabeled data. Furthermore, we denote the marginal distribution of feature $\boldsymbol{X}$ as $P_{\boldsymbol{X}}$ and its probability density function as $f_{\boldsymbol{X}}$. Similarly, $P_Y$ stands for the marginal distribution of target $Y$, and $f_Y$ is the density function of $P_Y$. We use $\mathbb{E}_{\boldsymbol{X},Y}, \mathbb{E}_{\boldsymbol{X}}$ and $\mathbb{E}_Y$ to denote the expectations over $P_{\boldsymbol{X},Y}, P_{\boldsymbol{X}}$, and $P_Y$, respectively.

Unlike Carpentier and Schlueter [2], we do not try to match unlabeled data and target values. In fact, our methods do not use each target value in $\mathcal{D}_Y$ but use density function $f_Y$ of the target, which can be estimated from $\mathcal{D}_Y$. For simplicity, we assume that the true density function $f_Y$ is known. The case where we need to estimate $f_Y$ from $\mathcal{D}_Y$ is discussed in Appendix B.

### 3.2    Pairwise comparison data

Here, we introduce pairwise comparison data. It consists of two random variables $(\boldsymbol{X}^+, \boldsymbol{X}^-)$, where the target value of $\boldsymbol{X}^+$ is larger than that of $\boldsymbol{X}^-$. Formally, $(\boldsymbol{X}^+, \boldsymbol{X}^-)$ are defined as

$$\boldsymbol{X}^+ = \begin{cases} \boldsymbol{X} & (Y \geq Y'), \\ \boldsymbol{X}' & (Y < Y'), \end{cases} \quad \boldsymbol{X}^- = \begin{cases} \boldsymbol{X}' & (Y \geq Y'), \\ \boldsymbol{X} & (Y < Y'), \end{cases} \tag{2}$$

where $(\boldsymbol{X}, Y), (\boldsymbol{X}', Y')$ are two independent pairs of random variables following $P_{\boldsymbol{X}, Y}$. We denote the joint distribution of $(\boldsymbol{X}^+, \boldsymbol{X}^-)$ as $P_{\boldsymbol{X}^+, \boldsymbol{X}^-}$ and the marginal distributions as $P_{\boldsymbol{X}^+}, P_{\boldsymbol{X}^-}$. Density functions $f_{\boldsymbol{X}^+, \boldsymbol{X}^-}, f_{\boldsymbol{X}^+}, f_{\boldsymbol{X}^-}$ and expectations $\mathbb{E}_{\boldsymbol{X}^+, \boldsymbol{X}^-}, \mathbb{E}_{\boldsymbol{X}^+}, \mathbb{E}_{\boldsymbol{X}^-}$ are defined in the same way.

We assume that we have access to $n_{\mathrm{R}}$ pairs of i.i.d. samples of $(\boldsymbol{X}^+, \boldsymbol{X}^-)$ as $\mathcal{D}_{\mathrm{R}} = \{(\boldsymbol{x}_i^+, \boldsymbol{x}_i^-)\}_{i=1}^{n_{\mathrm{R}}}$ in addition to unlabeled data $\mathcal{D}_{\mathrm{U}}$ and density function $f_Y$ of target variable $Y$. In the following sections, we show that uncoupled regression can be solved only from this information. In fact, our methods only require samples of *either one of* $\boldsymbol{X}^+, \boldsymbol{X}^-$, which corresponds to the case where only a winner or loser of the ranking is observable.

One naive approach to conducting uncouple regression with $\mathcal{D}_{\mathrm{R}}$ would be to adopt *ranking method*, which is to learn a ranker $r : \mathcal{X} \to \mathbb{R}$ that minimizes the following expected ranking loss:

$$R_{\mathrm{R}}(r) = \mathbb{E}_{\boldsymbol{X}^+, \boldsymbol{X}^-} \left[ \mathbb{1} \left[ r(\boldsymbol{X}^+) - r(\boldsymbol{X}^-) < 0 \right] \right], \tag{3}$$

where $\mathbb{1}$ is the indicator function. By minimizing the empirical estimation of (3) based on $\mathcal{D}_{\mathrm{R}}$, we can learn a ranker $\hat{r}$ that can sort data points by target $Y$. Then, we can predict quantiles of test data by ranking $\mathcal{D}_{\mathrm{U}}$, which leads to the prediction by applying the inverse of the cumulative distribution function (CDF) of $Y$. Formally, if the test point $\boldsymbol{x}_{\mathrm{test}}$ is ranked top $n'$-th in $\mathcal{D}_{\mathrm{U}}$, we can predict the target value for $\boldsymbol{x}_{\mathrm{test}}$ as

$$\hat{h}(\boldsymbol{x}_{\mathrm{test}}) = F_Y^{-1} \left( \frac{n_{\mathrm{U}} - n'}{n_{\mathrm{U}}} \right), \tag{4}$$

where $F_Y(t) = P(Y \le t)$ is the CDF of $Y$.

This approach, however, is known to be highly sensitive to the randomness in the target variable as discussed in Rigollet and Weed [28]. This is because a noise involved in the single data point changes the ranking of all other data points and affects their predictions. As illustrated in Rigollet and Weed [28], even if when we have a perfect ranker, i.e., we know the true order in $\mathcal{D}_{\mathrm{U}}$, model (4) is still different from the expected target $Y$ given feature $\boldsymbol{X}$ in presence of noise.

## 4 Empirical risk minimization by risk approximation

In this section, we propose a method to learn a model from pairwise comparison data $\mathcal{D}_{\mathrm{R}}$, unlabeled data $\mathcal{D}_{\mathrm{U}}$, and density function $f_Y$ of target variable $Y$. The method follows the empirical risk minimization principle, while the risk is approximated so that it can be empirically estimated from available data. Therefore, we call this method the *risk approximation* (RA) method. Here, we present an approximated risk and derive its estimation error bound.

From the definition of the Bregman divergence, the risk function in (1) is expressed as

$$R(h) = \mathbb{E}_Y \left[ \phi(Y) \right] - \mathbb{E}_{\boldsymbol{X}} \left[ \phi(h(\boldsymbol{X})) - h(\boldsymbol{X})\phi'(h(\boldsymbol{X})) \right] - \mathbb{E}_{\boldsymbol{X}, Y} \left[ Y\phi'(h(\boldsymbol{X})) \right]. \tag{5}$$

In this decomposition, the last term is the only problematic part in uncoupled regression since it requires to calculate the expectation on the joint distribution. Here, we consider approximating the last term based on the following expectations over the distributions of $\boldsymbol{X}^+, \boldsymbol{X}^-$.

**Lemma 1.** *We have*

$$\mathbb{E}_{\boldsymbol{X}^+} \left[ \phi'(h(\boldsymbol{X}^+)) \right] = 2\mathbb{E}_{\boldsymbol{X}, Y} \left[ F_Y(Y)\phi'(h(\boldsymbol{X})) \right],$$
$$\mathbb{E}_{\boldsymbol{X}^-} \left[ \phi'(h(\boldsymbol{X}^-)) \right] = 2\mathbb{E}_{\boldsymbol{X}, Y} \left[ (1 - F_Y(Y))\phi'(h(\boldsymbol{X})) \right].$$

The proof can be found in Appendix C.1. From Lemma 1, we can see that $\mathbb{E}_{\boldsymbol{X}, Y} \left[ Y\phi'(h(\boldsymbol{X})) \right] = (\mathbb{E}_{\boldsymbol{X}^+} \left[ \phi'(h(\boldsymbol{X}^+)) \right])/2$ if $F_Y(y) = y$, which corresponds to the case where target variable $Y$ marginally distributes uniformly in $[0, 1]$. This leads us to consider the approximation in the form of

$$\mathbb{E}_{\boldsymbol{X}, Y} \left[ Y\phi'(h(\boldsymbol{X})) \right] \simeq w_1 \mathbb{E}_{\boldsymbol{X}^+} \left[ \phi'(h(\boldsymbol{X}^+)) \right] + w_2 \mathbb{E}_{\boldsymbol{X}^-} \left[ \phi'(h(\boldsymbol{X}^-)) \right] \tag{6}$$

for some constants $w_1, w_2 \in \mathbb{R}$. Note that the above uniform case corresponds to $(w_1, w_2) = (1/2, 0)$. In general, if target $Y$ marginally distributes uniformly on $[a, b]$ for $b > a$, that is, $F_Y(y) = (y - a)/(b - a)$ for all $y \in [a, b]$, we can see that approximation (6) becomes exact for $(w_1, w_2) = (b/2, a/2)$ from Lemma 1. In such a case, we can construct an unbiased estimator of true risk $R$ from

unlabeled and pairwise comparison data. For non-uniform target marginal distributions, we choose $(w_1, w_2)$ that minimizes the upper bound of the estimation error, which we will discuss in detail later.

Since we have $\mathbb{E}_{\boldsymbol{X}}\left[\phi'(\boldsymbol{X})\right] = \frac{1}{2}\mathbb{E}_{\boldsymbol{X}^+}\left[\phi'(\boldsymbol{X}^+)\right] + \frac{1}{2}\mathbb{E}_{\boldsymbol{X}^-}\left[\phi'(\boldsymbol{X}^-)\right]$ from Lemma 1, (6) can be rewritten as

$$\mathbb{E}_{\boldsymbol{X},Y}\left[Y\phi'(h(\boldsymbol{X}))\right]$$
$$\simeq \lambda\mathbb{E}_{\boldsymbol{X}}\left[\phi'(\boldsymbol{X})\right] + \left(w_1 - \frac{\lambda}{2}\right)\mathbb{E}_{\boldsymbol{X}^+}\left[\phi'(h(\boldsymbol{X}^+))\right] + \left(w_2 - \frac{\lambda}{2}\right)\mathbb{E}_{\boldsymbol{X}^-}\left[\phi'(h(\boldsymbol{X}^-))\right] \quad (7)$$

for arbitrary $\lambda \in \mathbb{R}$. Hence, by approximating (5) by (7), we can write the approximated risk $R_{\mathrm{RA}}$ as

$$R_{\mathrm{RA}}(h; \lambda, w_1, w_2) = \mathfrak{C} - \mathbb{E}_{\boldsymbol{X}}\left[\phi(h(\boldsymbol{X})) - (h(\boldsymbol{X}) - \lambda)\phi'(h(\boldsymbol{X}))\right]$$
$$- \left(w_1 - \frac{\lambda}{2}\right)\mathbb{E}_{\boldsymbol{X}^+}\left[\phi'(h(\boldsymbol{X}^+))\right] - \left(w_2 - \frac{\lambda}{2}\right)\mathbb{E}_{\boldsymbol{X}^-}\left[\phi'(h(\boldsymbol{X}^-))\right].$$

Here, $\mathfrak{C} = \mathbb{E}_Y\left[\phi(Y)\right]$ can be ignored in the optimization procedure. Now, the empirical estimator of $R_{\mathrm{RA}}$ is

$$\hat{R}_{\mathrm{RA}}(h; \lambda, w_1, w_2) = \mathfrak{C} - \frac{1}{n_{\mathrm{U}}}\sum_{\boldsymbol{x}_i \in \mathcal{D}_{\mathrm{U}}}\left(\phi(h(\boldsymbol{x}_i)) - (h(\boldsymbol{x}_i) - \lambda)\phi'(h(\boldsymbol{x}_i))\right)$$
$$- \frac{1}{n_{\mathrm{R}}}\sum_{(\boldsymbol{x}_i^+, \boldsymbol{x}_i^-) \in \mathcal{D}_{\mathrm{R}}}\left(\left(w_1 - \frac{\lambda}{2}\right)\phi'(h(\boldsymbol{x}_i^+)) + \left(w_2 - \frac{\lambda}{2}\right)\phi'(h(\boldsymbol{x}_i^-))\right),$$

which is to be minimized in the RA method. Again, we would like to emphasize that if marginal distribution $P_Y$ is uniform on $[a, b]$ and $(w_1, w_2)$ is set to $(b/2, a/2)$, we have $R_{\mathrm{RA}} = R$ and $\hat{R}_{\mathrm{RA}}$ is an unbiased estimator of $R$ for any $\lambda \in \mathbb{R}$.

From the definition of $\hat{R}_{\mathrm{RA}}$, we can see that by setting $\lambda$ to either $2w_1$ or $2w_2$, $\hat{R}_{\mathrm{RA}}$ becomes independent of either $\boldsymbol{X}^+$ or $\boldsymbol{X}^-$. This means that we can conduct uncouple regression even if one of $\boldsymbol{X}^+, \boldsymbol{X}^-$ is missing in data, which corresponds to the case where only winners or only losers of the comparison are observed.

Another advantage of tuning free parameter $\lambda$ is that we can reduce the variance in empirical risk $\hat{R}_{\mathrm{RA}}$ as discussed in Sakai et al. [30] and Bao et al. [1]. As in Sakai et al. [30], the optimal $\lambda$ that minimizes the variance in $\hat{R}_{\mathrm{RA}}$ for $n_{\mathrm{U}} \to \infty$ is derived as follows.

**Theorem 1.** *For a given model $h$, let $\sigma_+^2, \sigma_-^2$ be*

$$\sigma_+^2 = \mathrm{Var}_{\boldsymbol{X}^+}\left[\phi'(h(\boldsymbol{X}^+))\right], \quad \sigma_-^2 = \mathrm{Var}_{\boldsymbol{X}^-}\left[\phi'(h(\boldsymbol{X}^-))\right],$$

*respectively, where $\mathrm{Var}_{\boldsymbol{X}}\left[\cdot\right]$ is the variance with respect to the random variable $\boldsymbol{X}$. Then, setting*

$$\lambda = \frac{2(w_1\sigma_+^2 + w_2\sigma_-^2)}{\sigma_+^2 + \sigma_-^2}$$

*yields the estimator with the minimum variance among estimators in the form of $\hat{R}_{\mathrm{RA}}$ when $n_{\mathrm{U}} \to \infty$.*

The proof can be found in Appendix C.3. From Theorem 1, we can see that the optimal $\lambda$ does not equal zero, which means that we can reduce the variance in the empirical estimation with a sufficient number of unlabeled data by tuning $\lambda$. Note that this situation is natural since unlabeled data is easier to collect than pairwise comparison data as discussed in Duh and Kirchhoff [9].

Now, from the discussion of the pseudo-dimension [12], we establish an upper bound of the following estimation error, which is used to choose weights $(w_1, w_2)$. Let $\hat{h}_{\mathrm{RA}}$ and $h^*$ be the minimizers of $\hat{R}_{\mathrm{RA}}$ and $R$ in hypothesis class $\mathcal{H}$, respectively. Then, we have the following theorem that bounds the excess risk in terms of parameters $(w_1, w_2)$.

**Theorem 2.** *Suppose that the pseudo-dimensions of $\{\boldsymbol{x} \to \phi'(h(\boldsymbol{x})) \mid h \in \mathcal{H}\}, \{\boldsymbol{x} \to h(\boldsymbol{x})\phi'(h(\boldsymbol{x})) - \phi(h(\boldsymbol{x})) \mid h \in \mathcal{H}\}$ are finite and there exist constants $m, M$ such that $|h(\boldsymbol{x})\phi'(h(\boldsymbol{x})) - \phi(h(\boldsymbol{x}))| \le m, |\phi'(h(\boldsymbol{x}))| \le M$ for all $\boldsymbol{x} \in \mathcal{X}$ and all $h \in \mathcal{H}$. Then,*

$$R(\hat{h}_{\mathrm{RA}}) \le R(h^*) + O\left(\sqrt{\frac{\log 1/\delta}{n_{\mathrm{U}}}}\right) + O\left(\sqrt{\frac{\log 1/\delta}{n_{\mathrm{R}}}}\right) + M\mathrm{Err}(w_1, w_2)$$

*holds with probability $1 - \delta$, where* Err *is defined as*

$$\text{Err}(w_1, w_2) = \mathbb{E}_Y \left[ |Y - 2w_1 F_Y(Y) - 2w_2(1 - F_Y(Y))| \right]. \tag{8}$$

The proof can be found in Appendix C.2. Note that the conditions for the boundedness of $|h(\boldsymbol{x})\phi'(h(\boldsymbol{x})) - \phi(h(\boldsymbol{x}))|, |\phi'(h(\boldsymbol{x}))|$ hold for many losses, e.g., the $l_2$-loss, when we consider a hypothesis space of bounded functions.

From Theorem 2, we can see that we can learn a model with less excess risk by minimizing $\text{Err}(w_1, w_2)$. Note that $\text{Err}(w_1, w_2)$ can be easily minimized since density function $f_Y$ is known or can be estimated from $\mathcal{D}_Y$. In particular, if target $Y$ is uniformly distributed on $[a, b]$, we have $\text{Err}(w_1, w_2) = 0$ by setting $(w_1, w_2) = (b/2, a/2)$. In such a case, $\hat{h}_{\text{RA}}$ becomes a consistent model, i.e., $R(\hat{h}_{\text{RA}}) \to R(h^*)$ as $n_{\text{U}}, n_{\text{R}} \to \infty$. The convergence rate is $O(1/\sqrt{n_{\text{U}}} + 1/\sqrt{n_{\text{R}}})$, which is the optimal parametric rate for the empirical risk minimization without additional assumptions when the enough amount of unlabeled and pairwise comparison data is provided jointly [21].

One important case where target variable $Y$ distributes uniformly is when the target is a "quantile value". For instance, we are to build a screening system for credit cards. Then, what we are interested in is "how much is an applicant credible in the population?", which means that we want to predict the quantile value of the "credit score" in the marginal distribution. By definition, we know that such a quantile value distributes uniformly, and thus we can have a consistent model by minimizing $\hat{R}_{\text{RA}}$.

In general cases, however, we may have $\text{Err}(w_1, w_2) > 0$, and $\hat{h}_{\text{RA}}$ becomes not consistent. Nevertheless, this is inevitable as suggested in the following theorem.

**Theorem 3.** *There exists a pair of joint distributions $P_{\boldsymbol{X},Y}, \tilde{P}_{\boldsymbol{X},Y}$ that yields the same marginal distributions of feature $P_{\boldsymbol{X}}$ and target $P_Y$, and the same distributions of the pairwise comparison data $P_{\boldsymbol{X}^+, \boldsymbol{X}^-}$ but have different conditional expectation $\mathbb{E}_{Y|\boldsymbol{X}=\boldsymbol{x}}[Y]$.*

Theorem 3 states that there exists a pair of distributions that cannot be distinguished from available data. Considering that $h^*(\boldsymbol{x}) = \mathbb{E}_{Y|\boldsymbol{X}=\boldsymbol{x}}[Y]$ when hypothesis space $\mathcal{H}$ is rich enough [11], this theorem implies that we cannot always obtain a consistent model. Still, in Section 6, we show that $h_{\text{RA}}$ empirically exhibits a similar accuracy to a model learned from ordinary coupled data.

## 5 Empirical risk minimization by target transformation

In this section, we introduce another method to uncoupled regression with pairwise comparison data, called the *target transformation* (TT) method. Whereas the RA method minimizes the approximation of the original risk, the TT method transforms the target variable so that it marginally distributes uniformly and minimizes an unbiased estimator of the risk defined based on the transformed variable.

Although there are several ways to map $Y$ to a uniformly distributed random variable, one natural candidate would be CDF $F_Y(Y)$, which leads to the following risk:

$$R_{\text{TT}}(h) = \mathbb{E}_{\boldsymbol{X},Y} \left[ d_\phi(F_Y(Y), F_Y(h(\boldsymbol{X}))) \right]. \tag{9}$$

Since $F_Y(Y)$ distributes uniformly on $[0, 1]$ by definition, we can construct the following unbiased estimator of $R_{\text{TT}}$ by the same discussion as in the previous section.

$$\hat{R}_{\text{TT}}(h; \lambda) = \mathfrak{C} - \frac{1}{n_{\text{U}}} \sum_{\boldsymbol{x}_i \in \mathcal{D}_{\text{U}}} \left( (\lambda - F_Y(h(\boldsymbol{x}_i)))\phi'(F_Y(h(\boldsymbol{x}_i))) + \phi(F_Y(h(\boldsymbol{x}_i))) \right)$$
$$- \frac{1}{n_{\text{R}}} \sum_{(\boldsymbol{x}_i^+, \boldsymbol{x}_i^-) \in \mathcal{D}_{\text{R}}} \left( \frac{1 - \lambda}{2} \phi'(F_Y(h(\boldsymbol{x}_i^+))) - \frac{\lambda}{2} \phi'(F_Y(h(\boldsymbol{x}_i^-))) \right),$$

where $\lambda$ is a hyper-parameter to be tuned. The TT method minimizes $\hat{R}_{\text{TT}}$ to learn a model. However, the learned model is, again, not always consistent in terms of original risk $R$. This is because, in rich enough hypothesis space $\mathcal{H}$, the minimizer $h_{\text{TT}} = F_Y^{-1}\left( \mathbb{E}_{Y|\boldsymbol{X}=\boldsymbol{x}}[F_Y(Y)] \right)$ of (9) is different from $\mathbb{E}_{Y|\boldsymbol{X}=\boldsymbol{x}}[Y]$, the minimizer of (1), unless target $Y$ distributes uniformly. Hence, for a non-uniform target, we cannot always obtain a consistent model. However, we can still derive an estimation error bound if $h_{\text{TT}} \in \mathcal{H}$ and target variable $Y$ is generated as

$$Y = h_{\text{true}}(\boldsymbol{X}) + \varepsilon, \tag{10}$$

where $h_{\text{true}} : \mathcal{X} \to \mathcal{Y}$ is the true target function and $\varepsilon$ is a zero-mean noise variable bounded in $[-\sigma, \sigma]$ for some constant $\sigma > 0$.

**Theorem 4.** *Assume that target variable $Y$ is generated by* (10) *and $h_{\text{TT}} \in \mathcal{H}$. If the pseudo-dimensions of $\{\boldsymbol{x} \to \phi'(F_Y(h(\boldsymbol{x})))| h \in \mathcal{H}\}, \{\boldsymbol{x} \to \phi'(F_Y(h(\boldsymbol{x})))| h \in \mathcal{H}\}$ are finite and there exist constants $P > p > 0$ such that $p \le f_Y(y) \le P$ for all $y \in \mathcal{Y}$, we have*

$$ R(\hat{h}_{\text{TT}}) \le R(h_{\text{true}}) + \left(\frac{P}{p}\sigma\right)^2 + O\left(\sqrt{\frac{\log 1/\delta}{n_{\text{U}}}}\right) + O\left(\sqrt{\frac{\log 1/\delta}{n_{\text{R}}}}\right) $$

*with probability $1 - \delta$ for $\phi(x) = x^2$, where $\hat{h}_{\text{TT}}$ is the minimizer of risk $\hat{R}_{\text{TT}}$ in $\mathcal{H}$.*

The proof can be found in Appendix C.5. From Theorem 4, we can see that $\hat{h}_{\text{TT}}$ is not necessarily consistent. Again, this is inevitable due to the same reason as the RA method. We can see that the error in the TT method is explicitly dependent on the noise, and thus, it is advantageous when the target contains less noise. In Section 6, we empirically compare these methods and show that which method is more suitable differs from case to case.

## 6 Experiments

In this section, we present the empirical performances of the proposed methods in experiments based on synthetic data and benchmark data. We show that our proposed methods outperform the naive method described in (4) and existing method [24]. Moreover, it is shown that our methods have a similar performance to a model learned from ordinary supervised learning with coupled data.

Before presenting the results, we describe the detailed procedure of experiments. In all experiments, we consider $l_2$-loss $l(z, t) = (z - t)^2$, which corresponds to setting $\phi(x) = x^2$ in Bregman divergence $d_\phi(t, z)$. The performance is also evaluated by the mean squared error (MSE) in the held-out test data. We repeat each experiment for 100 times and report the mean and the standard deviation. We employ hypothesis space of linear functions $\mathcal{H} = \{h(\boldsymbol{x}) = \boldsymbol{\theta}^\top \boldsymbol{x} \mid \boldsymbol{\theta} \in \mathbb{R}^d\}$ for the RA method. A slightly different hypothesis space $\mathcal{H}' = \{h(\boldsymbol{x}) = F_Y^{-1}(\sigma(\boldsymbol{\theta}^\top \boldsymbol{x})) \mid \boldsymbol{\theta} \in \mathbb{R}^d\}$ is employed for the TT method in order to simplify the loss, where $\sigma$ is logistic function $\sigma(x) = 1/(1 + \exp(-x))$. The procedure of hyper-parameter tuning in $R_{\text{RA}}$ and $R_{\text{TT}}$ can be found in Appendix A.

### 6.1 Comparison with baseline methods

We introduce two types of baseline methods here. One is a naive application of the ranking methods described in (4), in which we use SVMRank [13] as a ranking method. We use the linear kernel in SVMRank. The other is an ordinary supervised linear regression (LR), in which we fit a linear model using the true labels in unlabeled data $\mathcal{D}_{\text{U}}$. Note that LR does not use pairwise comparison data $\mathcal{D}_{\text{R}}$.

**Result for synthetic data.** First, we show the result for the synthetic data, in which we know the true marginal $P_Y$. We sample 5-dimensional unlabeled data $\mathcal{D}_{\text{U}}$ from normal distribution $\mathcal{N}(\boldsymbol{0}, I_d)$, where $I_d$ is the identity matrix. Then, we sample true unknown parameter $\boldsymbol{\theta}$ such that $\|\boldsymbol{\theta}\|_2 = 1$ uniformly at random. Target $Y$ is generated as $Y = \boldsymbol{\theta}^\top \boldsymbol{X} + \varepsilon$, where $\varepsilon$ is a noise following $\mathcal{N}(0, 0.01)$. Consequently, $P_Y$ corresponds to $\mathcal{N}(0, \sqrt{1.01})$, which is utilized in the proposed methods and the ranking baseline. The pairwise comparison data is generated by (2). We first sample two features $\boldsymbol{X}, \boldsymbol{X}'$ from $\mathcal{N}(\boldsymbol{0}, I_d)$, and then, compare them based on the target value $Y, Y'$ calculated by $Y = \boldsymbol{\theta}^\top \boldsymbol{X} + \varepsilon$. We fix $n_{\text{U}}$ to 100,000 and alter $n_{\text{R}}$ from 20 to 10,240 to see the change of performance with respect to the size of pairwise comparison data.

The results are presented in Figure 1. From this figure, we can see that with sufficient pairwise comparison data, the performance of our methods is significantly better than SVMRank baseline and close to LR. This is astonishing since LR uses the true label of $\mathcal{D}_{\text{U}}$, while our methods do not. In this experiment, the RA method consistently performs better than the TT method, though this is not universal as shown in the experiments on benchmark datasets.

Note that the TT method is unstable when the size of pairwise comparison data is small. We observed this phenomenon in all experiments. This is because we learn the quantile value when we minimize

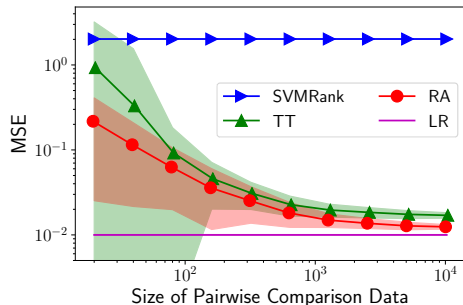

Figure 1: MSE for Synthetic Data

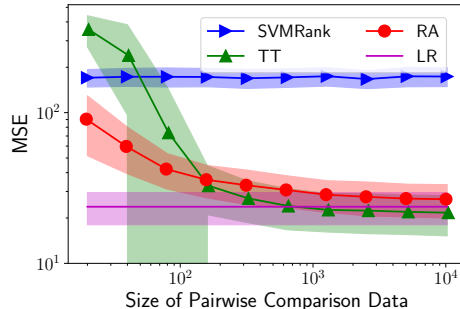

Figure 2: MSE for housing Dataset

Table 1: MSE for benchmark datasets when $n_R$ is 5,000. The bold face means the outstanding method in uncoupled regression methods (SVMRank, RA and TT) chosen by the Welch t-test with significance level 5%. Note that LR does not solve uncoupled regression since it uses labels in $\mathcal{D}_U$.

| | Supervised Regression | Uncoupled Regression | | |
|---|---|---|---|---|
| Dataset | LR | SVMRank | RA | TT |
| housing | 24.5(5.0) | 110.3(29.5) | 29.5(6.9) | **22.5(6.2)** |
| diabetes | 3041.9(219.8) | 8575.9(883.1) | **3087.3(256.3)** | **3127.3(278.8)** |
| airfoil | 23.3(2.2) | 62.1(7.6) | 23.7(2.0) | **22.7(2.2)** |
| concrete | 109.5(13.3) | 322.9(45.8) | **111.7(13.2)** | 139.1(17.9) |
| powerplant | 20.6(0.9) | 372.2(34.8) | **21.8(1.1)** | **22.0(1.0)** |
| mpg | 12.1(2.04) | 125(15.1) | 12.8(2.16) | **10.3(2.08)** |
| redwine | 0.412(0.0361) | 1.28(0.112) | **0.442(0.0473)** | 0.466(0.0412) |
| whitewine | 0.574(0.0325) | 1.58(0.0691) | **0.597(0.0382)** | 0.644(0.0414) |
| abalone | 5.05(0.375) | 20.9(1.44) | **5.26(0.372)** | 5.54(0.424) |

$R_{TT}$, and this can be severely inaccurate when the size of pairwise comparison data is small. On the other hand, $R_{RA}$ directly minimizes the approximation of true risk $R$, which is less sensitive to the size of $\mathcal{D}_R$.

**Result for benchmark datasets.** We conducted the experiments for the benchmark datasets as well, in which we do not know true marginal $P_Y$. The details of benchmark datasets can be found in Appendix A. We use the original features as unlabeled data $\mathcal{D}_U$. Density function $f_Y$ is estimated from target values in the dataset by kernel density estimation [25] with the Gaussian kernel. Here, the bandwidth of Gaussian kernel is determined by cross-validation. The pairwise comparison data is constructed by comparing the true target values of two data points uniformly sampled from $\mathcal{D}_U$.

Figure 2 shows[3] the performance of each method with respect to the size of pairwise comparison data for the housing dataset. We can see that the proposed methods significantly outperform SVMRank and approach to LR with increasing $n_R$. This fact suggests that the estimation error in $f_Y$ has little impact on the performance. The results for various datasets when $n_R$ is 5,000 are presented in Table 1, in which both proposed methods show the promising performance. Note that the method with less MSE differs by each dataset, which means that we cannot easily judge which method is better.

## 6.2 Comparison with other uncoupled regression methods

Here, we show the results of the empirical comparison between our methods and the method proposed in Pananjady et al. [24], which is another uncoupled regression method. Note that Pananjady et al. [24] considered a different problem, since they assume that the true regression function is exactly a linear function of the features and ignore the comparative data. Hence, we synthetically create data that all three methods are applicable and conduct comparison based on it.

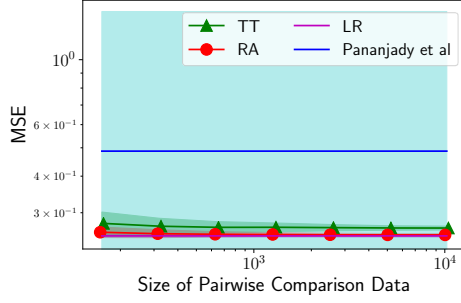
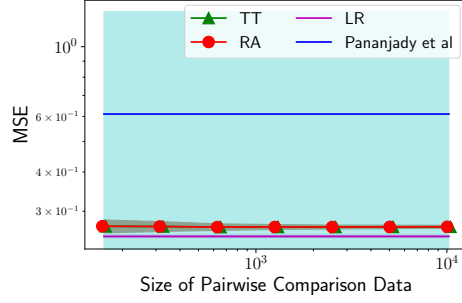

Figure 3: MSE for synthetic data following normal distribution

Figure 4: MSE for 1-dimensional synthetic data following uniform distribution

This method is to learn the optimal coupling of unlabeled data $\mathcal{D}_U = \{x_i\}_{i=1}^{n_U}$ and the target values $\mathcal{D}_Y = \{y_i\}_{i=1}^{n_U}$, assuming the linear relationship between them. Formally, the method is to find optimal parameter $\hat{\theta}$ and permutation $\hat{\sigma} : [n_U] \rightarrow [n_U]$ that minimizes the MSE:

$$(\hat{\boldsymbol{\theta}}, \hat{\sigma}) = \underset{\boldsymbol{\theta} \in \mathbb{R}^d, \sigma \in \mathcal{P}_n}{\arg\min} \sum_{i=1}^{n_U} (y_{\sigma(i)} - \boldsymbol{x}_i^\top \boldsymbol{\theta})^2,$$

where $\mathcal{P}_n$ is the set of permutations of $n$ items. Pananjady et al. [24] proved that this minimization is NP-hard in general but can be solved with $O(n_U \log n_U)$ computation when $d = 1$. Hence, we conduct the experiment based on synthetic 1-dimensional data.

The data is generated as follows. Unlabeled data $\mathcal{D}_U = \{x_i\}_{i=1}^{n_U}$ is sampled from a certain distribution, where the size of data is fixed as $n_U = 100,000$. Here, we used normal distribution $\mathcal{N}(0, 1)$ and uniform distribution on $[-1, 1]$. We set $\theta = -1$ and generate target $Y$ as $Y = X + \varepsilon$, where $\varepsilon$ is a noise following $\mathcal{N}(0, 0.25)$. We randomly shuffle target values $y_i$ to build $\mathcal{D}_Y$. The comparative data is constructed in the same way as the synthetic data described in Section 6.1. Note that the method in Pananjady et al. [24] ignores comparative data, hence the performance does not depend on the amount of comparative data.

The results[4] are shown in Figures 3 and 4, which show the superiority of our method. This is mainly due to the difference in data used in each method. The method in Pananjady et al. [24] only uses the unlabeled data and target values, while our methods utilize the comparative data as well. We can see that this additional information greatly contributes to better performance.

## 7 Conclusions

In this paper, we proposed novel methods for uncoupled regression by utilizing pairwise comparison data. We introduced two methods, the risk approximation (RA) method, and the target transformation (TT) method, for the problem. The RA method is to approximate the expected Bregman divergence by the linear combination of expectations of given data, and the TT method is to learn a model for quantile values and uses the inverse of the CDF to predict the target. We derived estimation error bounds for each method and showed that the learned model is consistent when the target variable distributes uniformly. Furthermore, the empirical evaluations based on both synthetic data and benchmark datasets suggested the competence of our method. The empirical results also indicated the instability of the TT method when the size of pairwise comparison data is small, and we may need some regularization scheme to prevent it, which is left for future work.

**Acknowledgements**

LX utilized the facility provided by Masason Foundation. JH acknowledges support by KAKENHI 18K17998, and MS was supported by JST CREST Grant Number JPMJCR18A2.

## Footnotes

*Now at Gatsby Computational Neuroscience Unit

[2]This questioning can be regarded as one type of randomized response (indirect questioning) techniques [32], which is a survey method to avoid social desirability bias.

[3]From Figure 2, we can again see that the TT method performs unstably when $n_R$ is small for benchmark data, which causes the strange profile in the log plot. Since the standard deviation of the TT method is large, the mean accuracy minus the standard deviation goes negative, which diverges to $-\infty$ in the plot.

[4]The standard deviation of the method in Pananjady et al. [24] is large but finite. The mean minus standard deviation diverges to $-\infty$ in the plot for the same reason as Figure 2.

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
