[Supplementary Material]

Table 2: Information of benchmark datasets.

| Dataset | Datasize | $d$ | Source |
|---------|----------|-----|--------|
| housing | 404 | 13 | UCI Repository |
| airfoil | 1202 | 5 | UCI Repository |
| concrete | 824 | 8 | UCI Repository |
| powerplant | 7654 | 4 | UCI Repository |
| mpg | 313 | 7 | UCI Repository |
| redwine | 1279 | 11 | UCI Repository |
| whitewine | 3918 | 11 | UCI Repository |
| abalone | 3341 | 10 | UCI Repository |
| diabetes | 353 | 10 | [10] |

# A    Experiments details

In this appendix, we explain the detailed setting of experiments. First, we describe the procedure of hyper-parameter tuning during the experiments. Then, we provide detailed information on benchmark datasets.

## A.1    Procedure of hyper-parameter tuning

To construct risk $\hat{R}_{\mathrm{RA}}$, we need to tune $\lambda, w_1, w_2$, which is done by minimizing empirically approximated $\mathrm{Err}(w_1, w_2)$ defined in (8). Let $\overline{y}, \underline{y}$ be the 0.99-quantile and 0.01-quantile of $P_Y$, respectively. Note that we can calculate these quantities since we have access to $f_Y$. Then, we define $\{y^{(i)}\}_{i=1}^{n_{\mathrm{split}}+1}$ as $y_i = \underline{y} + (i-1)/n_{\mathrm{split}}(\overline{y} - \underline{y})$, by which $\mathrm{Err}(w_1, w_2)$ is approximated as

$$\mathrm{Err}(w_1, w_2) \simeq \sum_{i=1}^{n_{\mathrm{split}}+1} f_Y(y_i)|y_i - w_1 F_Y(y_i) - w_2(1 - F_Y(y_i))|.$$

We employ $w_1, w_2$ that minimize the empirical approximation above with $n_{\mathrm{split}} = 1000$ and fix $\lambda$ to be $(w_1 + w_2)/2$ in all cases.

For the TT method, we employ hypothesis space $\mathcal{H}' = \{h(\boldsymbol{x}) = F_Y^{-1}(\sigma(\boldsymbol{\theta}^\top \boldsymbol{x})) \mid \boldsymbol{\theta} \in \mathbb{R}^d\}$, which is slightly different from hypothesis space of liner functions $\mathcal{H} = \{h(\boldsymbol{x}) = \boldsymbol{\theta}^\top \boldsymbol{x} \mid \boldsymbol{\theta} \in \mathbb{R}^d\}$, where $\sigma$ is logistic function $\sigma(x) = 1/(1 + \exp(-x))$ This simplifies the loss function and reduces the computational time. We fix $\lambda = 1/2$ for this risk, which yields the loss

$$\hat{R}_{\mathrm{TT}}(h) = \mathfrak{C} - \frac{1}{n_{\mathrm{U}}} \sum_{\boldsymbol{x}_i \in \mathcal{D}_{\mathrm{U}}} \left(\frac{1}{2} - \sigma(\boldsymbol{\theta}^\top \boldsymbol{x}_i)\right) \phi'(\sigma(h(\boldsymbol{x}_i))) + \phi(\sigma(\boldsymbol{\theta}^\top \boldsymbol{x}_i))$$

$$- \frac{1}{n_{\mathrm{R}}} \sum_{(\boldsymbol{x}_i^+, \boldsymbol{x}_i^-) \in \mathcal{D}_{\mathrm{R}}} \frac{1}{4} \phi'(\sigma(\boldsymbol{\theta}^\top \boldsymbol{x}_i^+)) - \frac{1}{4} \phi'(\sigma(\boldsymbol{\theta}^\top \boldsymbol{x}_i^-)).$$

We minimize this loss with respect to $\boldsymbol{\theta}$.

## A.2    Benchmark dataset details

We use eight benchmark datasets from UCI repository [8] and one (diabetes) from Efron et al. [10]. The details of datasets can be found in Table 2. As preprocessing, we excluded all instances that contain missing values, and we encoded a categorical feature in abalone as a one-hot vector.

# B    Estimating density function and cumulative distribution function

In this section, we discuss the case where the true probability density function $f_Y$ is not given. In such a case, we need a slight modification of proposed methods since we have to estimate $f_Y$ from the set of target values $\mathcal{D}_Y = \{y_i\}_{i=1}^{n_Y}$, where $n_Y$ is the size of $\mathcal{D}_Y$. We first introduce a modification of the RA method and derive an estimation error bound for it. Then, we discuss the same for the TT method as well.

## B.1 Modification of the risk approximation method

Although $\hat{R}_{\mathrm{RA}}$ does not depend on $f_Y$ or $F_Y$, we need the information of $P_Y$ when tuning weights $w_1, w_2$, which is done by the minimization of Err defined in (8). Since, Err can not be directly calculated without $f_Y$ and $F_Y$, we propose another quantity $\widehat{\mathrm{Err}}$ below, which substitute expectation over $P_Y$ and CDF function $F_Y$ to empirical mean and the empirical CDF.

$$\widehat{\mathrm{Err}}(w_1, w_2) = \frac{1}{n_Y} \sum_{i=1}^{n_Y} |y_i - w_1 \hat{F}_Y(y_i) - w_2(1 - \hat{F}_Y(y_i))|,$$

where $\hat{F}_Y$ is the empirical CDF defined as

$$\hat{F}_Y(y) = \frac{1}{n_Y} \sum_{i=1}^{n_Y} \mathbb{1}\left[y_i \leq y\right].$$

Note that $\widehat{\mathrm{Err}}$ can be minimized given $\mathcal{D}_Y$. To show the validity of the method, we establish an estimation error bound involving $\widehat{\mathrm{Err}}$ as follows.

**Theorem 5.** *Let $\mathcal{Y}$ be bounded in $\mathcal{Y} \subseteq [-L, L]$. Then, for all $w_1, w_2 \in [-L, L]$, we have*

$$|\mathrm{Err}(w_1, w_2) - \widehat{\mathrm{Err}}(w_1, w_2)| \leq O\left(\sqrt{\frac{\log \delta}{n_Y}}\right)$$

*with probability $1 - 2\delta$.*

*Proof.* Since the weights are bounded, from Mohri et al. [22, Thm. 10.3], we have

$$\mathrm{Err}(w_1, w_2) \leq \frac{1}{n_Y} \sum_{i=1}^{n_Y} |y_i - w_1 F_Y(y_i) - w_2(1 - F_Y(y_i))| + O\left(\sqrt{\frac{\log 1/\delta}{m}}\right),$$

with probability $1 - \delta$. Furthermore, from Dvoretzky-Kiefer-Wolfowitz inequality [20], we have

$$\|F_Y(y) - \hat{F}_Y(y)\|_\infty \leq \sqrt{\frac{\log(2/\delta)}{2n_Y}} \tag{11}$$

with probability $1 - \delta$, which yields

$$\frac{1}{n_Y} \sum_{i=1}^{n_Y} |y_i - w_1 F_Y(y_i) - w_2(1 - F_Y(y_i))| \leq \widehat{\mathrm{Err}} + O\left(\sqrt{\frac{\log 1/\delta}{m}}\right).$$

Therefore, from the union bound, we have

$$|\mathrm{Err}(w_1, w_2) - \widehat{\mathrm{Err}}(w_1, w_2)| \leq O\left(\sqrt{\frac{\log \delta}{n_Y}}\right)$$

with probability $1 - 2\delta$. □

From Theorems 2 and 5, we have

$$R(\hat{h}_{\mathrm{RA}}) \leq R(h^*) + O\left(\sqrt{\frac{\log 1/\delta}{n_U}}\right) + O\left(\sqrt{\frac{\log 1/\delta}{n_R}}\right) + O\left(\sqrt{\frac{\log 1/\delta}{n_Y}}\right) + M\widehat{\mathrm{Err}}(w_1, w_2),$$

with probability $1 - 5\delta$ under the conditions given in these theorems.

## B.2 Modification on the target transformation method

Let $\tilde{R}_{\mathrm{TT}}$ be the risk which substitute $F_Y$ in $R_{\mathrm{TT}}$ to empirical CDF, defined as

$$\tilde{R}_{\mathrm{TT}}(h; \lambda) = \mathfrak{C} - \frac{1}{n_{\mathrm{U}}} \sum_{\boldsymbol{x}_i \in \mathcal{D}_{\mathrm{U}}} \left( (\lambda - \hat{F}_Y(h(\boldsymbol{x}_i)))\phi'(\hat{F}_Y(h(\boldsymbol{x}_i))) + \phi(\hat{F}_Y(h(\boldsymbol{x}_i))) \right)$$

$$- \frac{1}{n_{\mathrm{R}}} \sum_{(\boldsymbol{x}_i^+, \boldsymbol{x}_i^-) \in \mathcal{D}_{\mathrm{R}}} \left( \frac{1 - \lambda}{2} \phi'(\hat{F}_Y(h(\boldsymbol{x}_i^+))) - \frac{\lambda}{2} \phi'(\hat{F}_Y(h(\boldsymbol{x}_i^-))), \right).$$

Using (11), we have

$$|\hat{R}_{\mathrm{TT}}(h) - \tilde{R}_{\mathrm{TT}}(h)| \leq O\left( \sqrt{\frac{\log 1/\delta}{n_Y}} \right)$$

for all $h \in \mathcal{H}$ with probability $1 - \delta$. Let $\tilde{h}_{\mathrm{TT}}$ be the minimizer of $\tilde{R}_{\mathrm{TT}}$ in hypothesis space $\mathcal{H}$. Then, under the condition given in Theorem 4, we have

$$R_{\mathrm{TT}}(\tilde{h}_{\mathrm{TT}}) \leq R_{\mathrm{TT}}(h_{\mathrm{TT}}) + O\left( \sqrt{\frac{\log 1/\delta}{n_Y}} \right) + O\left( \sqrt{\frac{\log 1/\delta}{n_{\mathrm{R}}}} \right) + O\left( \sqrt{\frac{\log 1/\delta}{n_{\mathrm{U}}}} \right),$$

with probability $1 - 4\delta$, therefore we have

$$R(\tilde{h}_{\mathrm{TT}}) \leq R(h^*) + 2\left( \frac{P}{p}\sigma \right)^2 + O\left( \sqrt{\frac{\log 1/\delta}{n_Y}} \right) + O\left( \sqrt{\frac{\log 1/\delta}{n_{\mathrm{R}}}} \right) + O\left( \sqrt{\frac{\log 1/\delta}{n_{\mathrm{U}}}} \right)$$

with probability $1 - 4\delta$, which can be shown by the slight modification of the proof of Theorem 4.

## C  Proofs

### C.1  Proof of Lemma 1

Lemma 1 can be proved as follows.

*Proof of Lemma 1.* Let $f_{\boldsymbol{X}^+}$ be the probability density function (PDF) of $P_{\boldsymbol{X}^+}$. From the definition of $\boldsymbol{X}^+$, we have

$$f_{\boldsymbol{X}^+}(\boldsymbol{x}) = \frac{1}{Z} \iiint f_{\boldsymbol{X},Y}(\boldsymbol{x}, y) f_{\boldsymbol{X},Y}(\boldsymbol{x}', y') \mathbb{1}\left[ y > y' \right] \mathrm{d}y \mathrm{d}y' \mathrm{d}\boldsymbol{x}'$$

$$= \frac{1}{Z} \int f_{\boldsymbol{X},Y}(\boldsymbol{x}, y) \left[ \int f_Y(y') \mathbb{1}\left[ y > y' \right] \mathrm{d}y' \right] \mathrm{d}y$$

$$= \frac{1}{Z} \int f_{\boldsymbol{X},Y}(\boldsymbol{x}, y) F_Y(y) \mathrm{d}y,$$

where $Z$ is the normalizing constant and $f_{\boldsymbol{X},Y}(y)$ is the PDF of $P_{\boldsymbol{X},Y}$. Now, $Z$ is calculated as

$$Z = \iint f_{\boldsymbol{X},Y}(\boldsymbol{x}, y) F_Y(y) \mathrm{d}y \mathrm{d}\boldsymbol{x}$$

$$= \int f_Y(y) F_Y(y) \mathrm{d}y$$

$$= \frac{1}{2},$$

where the last equality holds from the integration by parts. Therefore, we have

$$\mathbb{E}_{\boldsymbol{X}^+} \left[ \phi'(\boldsymbol{X}^+) \right] = \int f_{\boldsymbol{X}^+}(\boldsymbol{x}) \phi'(\boldsymbol{x}) \mathrm{d}\boldsymbol{x}$$

$$= \int 2 \left\{ \int f_{\boldsymbol{X},Y}(\boldsymbol{x}, y) F_Y(y) \mathrm{d}y \right\} \phi'(\boldsymbol{x}) \mathrm{d}\boldsymbol{x}$$

$$= \mathbb{E}_{\boldsymbol{X},Y} \left[ F_Y(Y) \phi'(\boldsymbol{x}) \right].$$

The expectation over $P_{\boldsymbol{X}^-}$ can be derived in the same way. $\square$

## C.2 Proof of Theorem 2

Here, we show the proof of Theorem 2. First, we show the gap between $R$ and $R_{\mathrm{RA}}$ can be bounded as follows.

**Lemma 2.** *For all $h \in \mathcal{H}$, such that $|\phi'(h(\boldsymbol{x}))| \le M$ for all $\boldsymbol{x} \in \mathcal{X}$, we have*

$$|R(h) - R_{\mathrm{RA}}(h; \lambda; w_1, w_2)| \le M\mathrm{Err}(w_1, w_2)$$

*for all $\lambda \in \mathbb{R}$.*

*Proof.* From Lemma 1 and the fact $\mathbb{E}_{\boldsymbol{X}}\left[\phi'(\boldsymbol{X})\right] = \frac{1}{2}\mathbb{E}_{\boldsymbol{X}^+}\left[\phi'(\boldsymbol{X}^+)\right] + \frac{1}{2}\mathbb{E}_{\boldsymbol{X}^-}\left[\phi'(\boldsymbol{X}^-)\right]$, we have

$$
\begin{aligned}
&|R(h) - R_{\mathrm{RA}}(h; \lambda, w_1, w_2)| \\
&= \left|\mathbb{E}_{\boldsymbol{X},Y}\left[Y\phi'(h(\boldsymbol{X}))\right] - w_1\mathbb{E}_{\boldsymbol{X}^+}\left[\phi'(h(\boldsymbol{X}^+))\right] - w_2\mathbb{E}_{\boldsymbol{X}^-}\left[\phi'(h(\boldsymbol{X}^-))\right]\right| \\
&= \left|\int f_{\boldsymbol{X},Y}(\boldsymbol{x}, y)\phi'(h(\boldsymbol{x}))\{y - 2w_1 F_Y(y) - 2w_2(1 - F_Y(y))\}\mathrm{d}y\mathrm{d}\boldsymbol{x}\right| \\
&\le \int f_{\boldsymbol{X},Y}(\boldsymbol{x}, y)\left|\phi'(h(\boldsymbol{x}))\right|\left|y - 2w_1 F_Y(y) - 2w_2(1 - F_Y(y))\right|\mathrm{d}y\mathrm{d}\boldsymbol{x} \\
&\le M\int f_Y(y)\left|y - 2w_1 F_Y(y) - 2w_2(1 - F_Y(y))\right|\mathrm{d}y \\
&\le M\mathrm{Err}(w_1, w_2).
\end{aligned}
$$

$\square$

Now, Theorem 2 can be derived as follows.

*Proof of Theorem 2.* Let $\tilde{d}, \tilde{d}'$ be the pseudo-dimensions defined as

$$
\begin{aligned}
\tilde{d} &= \mathrm{Pdim}(\{\boldsymbol{x} \to \phi'(h(\boldsymbol{x})) \mid h \in \mathcal{H}\}), \\
\tilde{d}' &= \mathrm{Pdim}(\{\boldsymbol{x} \to h(\boldsymbol{x})\phi'(h(\boldsymbol{x})) - \phi(h(\boldsymbol{x})) \mid h \in \mathcal{H}\}),
\end{aligned}
$$

where $\mathrm{Pdim}(\mathcal{F})$ denotes the pseudo-dimension of the functional space $\mathcal{F}$. From the assumptions in Theorem 2, using the discussion in Mohri et al. [22, Theorem 10.6], each of following bound holds with probability $1 - \delta$ for all $h \in \mathcal{H}$.

$$
\left|\mathbb{E}_{\boldsymbol{X}^+}\left[\phi'(h(\boldsymbol{X}^+))\right] - \frac{1}{n_{\mathrm{R}}}\sum_{\boldsymbol{x}_i^+ \in \mathcal{D}_{\mathrm{R}}^+}\phi'(h(\boldsymbol{x}_i^+))\right| \le M\sqrt{\frac{2\tilde{d}\log\frac{en_{\mathrm{R}}}{\tilde{d}}}{n_{\mathrm{R}}}} + M\sqrt{\frac{\log\frac{1}{\delta}}{2n_{\mathrm{R}}}},
$$

$$
\left|\mathbb{E}_{\boldsymbol{X}^-}\left[\phi'(h(\boldsymbol{X}^-))\right] - \frac{1}{n_{\mathrm{R}}}\sum_{\boldsymbol{x}_i^- \in \mathcal{D}_{\mathrm{R}}^-}\phi'(h(\boldsymbol{x}_i^-))\right| \le M\sqrt{\frac{2\tilde{d}\log\frac{en_{\mathrm{R}}}{\tilde{d}}}{n_{\mathrm{R}}}} + M\sqrt{\frac{\log\frac{1}{\delta}}{2n_{\mathrm{R}}}},
$$

$$
\left|\mathbb{E}_{\boldsymbol{X}}\left[g(\boldsymbol{X})\right] - \frac{1}{n_{\mathrm{U}}}\sum_{\boldsymbol{x}_i \in \mathcal{D}_{\mathrm{U}}^+}g(\boldsymbol{x}_i)\right| \le m\sqrt{\frac{2\tilde{d}'\log\frac{en_{\mathrm{U}}}{\tilde{d}'}}{n_{\mathrm{U}}}} + m\sqrt{\frac{\log\frac{1}{\delta}}{2n_{\mathrm{U}}}},
$$

where $g(\boldsymbol{x}) = h(\boldsymbol{x})\phi'(h(\boldsymbol{x})) + \phi(h(\boldsymbol{x}))$. From the uniform bound, we have

$$
\begin{aligned}
&|R_{\mathrm{RA}}(h; w_1, w_2) - \hat{R}_{\mathrm{RA}}(h; \lambda, w_1, w_2)| \\
&\le \left(\left|w_1 - \frac{\lambda}{2}\right| + \left|w_2 - \frac{\lambda}{2}\right|\right)\left(M\sqrt{\frac{2\tilde{d}\log\frac{en_{\mathrm{R}}}{\tilde{d}}}{n_{\mathrm{R}}}} + M\sqrt{\frac{\log\frac{1}{\delta}}{2n_{\mathrm{R}}}}\right) \\
&\quad + (m + \lambda M)\left(\sqrt{\frac{2\tilde{d}'\log\frac{en_{\mathrm{U}}}{\tilde{d}'}}{n_{\mathrm{U}}}} + \sqrt{\frac{\log\frac{1}{\delta}}{2n_{\mathrm{U}}}}\right)
\end{aligned}
$$

with probability $1 - 3\delta$ for all $h \in \mathcal{H}$. Hence, with probability $1 - 3\delta$, we have

$$
\begin{aligned}
R(\hat{h}_{\mathrm{RA}}) &- R(h^*) \\
&\leq R_{\mathrm{RA}}(\hat{h}_{\mathrm{RA}}; \lambda, w_1, w_2) - R_{\mathrm{RA}}(h^*; \lambda, w_1, w_2) + |R(h^*) - R_{\mathrm{RA}}(h^*; \lambda, w_1, w_2)| \\
&\quad + |R(\hat{h}_{\mathrm{RA}}) - R_{\mathrm{RA}}(\hat{h}_{\mathrm{RA}}; \lambda, w_1, w_2)| \\
&\leq (R_{\mathrm{RA}}(\hat{h}_{\mathrm{RA}}; \lambda, w_1, w_2) - \hat{R}_{\mathrm{RA}}(h^*; \lambda, w_1, w_2)) \\
&\quad - (R_{\mathrm{RA}}(h^*; \lambda, w_1, w_2) - \hat{R}_{\mathrm{RA}}(h^*; \lambda, w_1, w_2)) + 2M\mathrm{Err}(w_1, w_2) \\
&\leq (R_{\mathrm{RA}}(\hat{h}_{\mathrm{RA}}; \lambda, w_1, w_2) - \hat{R}_{\mathrm{RA}}(\hat{h}_{\mathrm{RA}}; \lambda, w_1, w_2)) \\
&\quad - (R_{\mathrm{RA}}(h^*; , \lambda, w_1, w_2) - \hat{R}_{\mathrm{RA}}(h^*; \lambda, w_1, w_2)) + 2M\mathrm{Err}(w_1, w_2) \\
&\leq O\left(\sqrt{\frac{\log 1/\delta}{n_{\mathrm{U}}}}\right) + O\left(\sqrt{\frac{\log 1/\delta}{n_{\mathrm{R}}}}\right) + 2M\mathrm{Err}(w_1, w_2),
\end{aligned}
$$

where the second inequality holds from the fact $\hat{R}_{\mathrm{RA}}(\hat{h}_{\mathrm{RA}}; \lambda, w_1, w_2) \leq \hat{R}_{\mathrm{RA}}(\hat{h}^*; \lambda, w_1, w_2)$ and Lemma 2. □

## C.3 Proof of Theorem 1

Theorem 1 can be shown as follows.

*Proof of Theorem 1.* The variance of $\hat{R}_{\mathrm{RA}}$ denoted as $\mathrm{Var}\left[\hat{R}_{\mathrm{RA}}(h; \lambda, w_1, w_2)\right]$ can be expressed as

$$
\mathrm{Var}\left[\hat{R}_{\mathrm{RA}}(h; \lambda, w_1, w_2)\right] = \left(w_1 - \frac{\lambda}{2}\right)^2 \frac{\sigma_+^2}{n_{\mathrm{R}}} + \left(w_2 - \frac{\lambda}{2}\right)^2 \frac{\sigma_-^2}{n_{\mathrm{R}}}
$$

when $n_{\mathrm{U}} \to \infty$. By solving the above quadratic optimization problem, we have

$$
\arg \min_{\lambda} \mathrm{Var}\left[\hat{R}_{\mathrm{RA}}(h; \lambda, w_1, w_2)\right] = \frac{2(w_1 \sigma_+^2 + w_2 \sigma_-^2)}{\sigma_+^2 + \sigma_-^2}.
$$

□

## C.4 Proof of Theorem 3

We can construct a simple example satisfies the conditions in Theorem 3 as follows.

*Proof.* Let $f_{\boldsymbol{X}, Y}, \tilde{f}_{\boldsymbol{X}, Y}$ be the PDF of $P_{\boldsymbol{X}, Y}, \tilde{P}_{\boldsymbol{X}, Y}$, respectively. If we consider $\mathcal{X} = [-1, 1]$ and $\mathcal{Y} = [0, 4]$ and these PDF to be

$$
f_{\boldsymbol{X}, Y}(x, y) = \begin{cases} \frac{1}{6} & (y \in [0, 2] \cup [3, 4]), \\ 0 & (\text{otherwise}), \end{cases}
$$

$$
\tilde{f}_{\boldsymbol{X}, Y}(x, y) = \begin{cases} \frac{1}{8} & (x \in [-1, 0), y \in [0, 1)), \\ \frac{1}{4} & (x \in [-1, 0), y \in [1, 2)), \\ \frac{1}{8} & (x \in [-1, 0), y \in [3, 4]), \\ \frac{5}{24} & (x \in [0, 1], y \in [0, 1)), \\ \frac{1}{12} & (x \in [0, 1], y \in [1, 2)), \\ \frac{5}{24} & (x \in [0, 1], y \in [3, 4]), \\ 0 & (\text{otherwise}). \end{cases}
$$

Then, by the simple calculation, we can see that they have the same PDF $f_{\boldsymbol{X}}(x), f_Y(y), f_{\boldsymbol{X}^+, \boldsymbol{X}^-}(x^+, x^-)$, each represents the PDF of $P_{\boldsymbol{X}}, P_Y, P_{\boldsymbol{X}^+, \boldsymbol{X}^-}$, respectively, which are

$$
f_{\boldsymbol{X}}(x) = 0.5,
$$

$$
f_Y(y) = \begin{cases} \frac{1}{3} & (y \in [0, 2] \cup [3, 4]), \\ 0 & (\text{otherwise}), \end{cases}
$$

$$
f_{\boldsymbol{X}^+, \boldsymbol{X}^-}(x^+, x^-) = 0.25.
$$

However, the conditional expectation $\mathbb{E}_{Y|\boldsymbol{X}=x}[Y]$ defined on $P_{\boldsymbol{X},Y}$ is

$$\mathbb{E}_{Y|\boldsymbol{X}=x}[Y] = \frac{11}{6},$$

while the conditional expectation $\tilde{\mathbb{E}}_{Y|\boldsymbol{X}=x}[Y]$ defined on $\tilde{P}_{\boldsymbol{X},Y}$ is

$$\tilde{\mathbb{E}}_{Y|\boldsymbol{X}=x}[Y] = \begin{cases} \frac{7}{4} & (x \in [-1,0)), \\ \frac{23}{12} & (x \in [0,1]). \end{cases}$$

$\square$

## C.5 Proof of Theorem 4

The Theorem 4 can be shown as follows.

*Proof of Theorem 4.* We first show that under the conditions, we have

$$\|h_{\text{true}}(\boldsymbol{x}) - h_{\text{TT}}(\boldsymbol{x})\|_{\infty} \leq \frac{\sigma P}{p}.$$

Since $(F_Y(y))' = f_Y(y) \leq P$ and $(F_Y^{-1}(q))' = 1/f_Y(F^{-1}(q)) \leq 1/p$ for any $y \in \mathcal{Y}$ and $q \in [0,1]$, $F_Y(y), F_Y^{-1}(q)$ are $P, 1/p$-Lipschitz continuous, respectively. Therefore, we have

$$\begin{aligned} h_{\text{TT}}(\boldsymbol{x}) &= F_Y^{-1}(\mathbb{E}_{Y|\boldsymbol{X}=x}[F_Y(Y)]) \\ &= F_Y^{-1}(\mathbb{E}_{\epsilon}[F_Y(h_{\text{true}}(\boldsymbol{x}) + \varepsilon)]) \\ &\leq F_Y^{-1}(F_Y(h_{\text{true}}(\boldsymbol{x}) + \sigma P)) \\ &\leq h_{\text{true}}(\boldsymbol{x}) + \frac{\sigma P}{p} \end{aligned}$$

for all $\boldsymbol{x} \in \mathcal{X}$. With the same discussion, we have $|h_{\text{TT}}(\boldsymbol{x}) - h_{\text{true}}(\boldsymbol{x})| \leq \frac{\sigma P}{p}$. Therefore, we have

$$\|h_{\text{true}}(\boldsymbol{x}) - h_{\text{TT}}(\boldsymbol{x})\|_{\infty} \leq \frac{\sigma P}{p}.$$

Now, if $\phi(x) = x^2$, which means $R(h) = \mathbb{E}_{\boldsymbol{X},Y}[(h(\boldsymbol{X}) - Y)^2]$, we have

$$\begin{aligned} R(\hat{h}_{\text{TT}}) &= \mathbb{E}_{\boldsymbol{X},Y}\left[(\hat{h}_{\text{TT}}(\boldsymbol{x}) - Y)^2\right] \\ &= \mathbb{E}_{\boldsymbol{X},Y}\left[(\hat{h}_{\text{TT}}(\boldsymbol{x}) - \hat{h}_{\text{true}}(\boldsymbol{x}) + \hat{h}_{\text{true}}(\boldsymbol{x}) - Y)^2\right] \\ &= \mathbb{E}_{\boldsymbol{X}}\left[(\hat{h}_{\text{TT}}(\boldsymbol{x}) - \hat{h}_{\text{true}}(\boldsymbol{x}))^2\right] + \mathbb{E}_{\boldsymbol{X},Y}\left[(\hat{h}_{\text{true}}(\boldsymbol{x}) - Y)^2\right] \\ &\quad + 2\mathbb{E}_{\boldsymbol{X},Y}\left[(\hat{h}_{\text{TT}}(\boldsymbol{x}) - \hat{h}_{\text{true}}(\boldsymbol{x}))(\hat{h}_{\text{true}}(\boldsymbol{x}) - Y)\right] \\ &= \mathbb{E}_{\boldsymbol{X}}\left[(\hat{h}_{\text{TT}}(\boldsymbol{X}) - h_{\text{true}}(\boldsymbol{X}))^2\right] + \mathbb{E}_{\boldsymbol{X},Y}\left[(h_{\text{true}}(\boldsymbol{X}) - Y)^2\right] \\ &\leq R(h_{\text{true}}) + 2\mathbb{E}_{\boldsymbol{X}}\left[(\hat{h}_{\text{TT}}(\boldsymbol{x}) - h_{\text{TT}}(\boldsymbol{x}))^2\right] + 2\mathbb{E}_{\boldsymbol{X}}\left[(h_{\text{true}}(\boldsymbol{x}) - h_{\text{TT}}(\boldsymbol{x}))^2\right]. \end{aligned}$$

Since $\|h_{\text{TT}}(\boldsymbol{x}) - h_{\text{true}}(\boldsymbol{X})\|_{\infty} \leq \frac{\sigma P}{p}$, we have

$$\mathbb{E}_{\boldsymbol{X}}\left[(h_{\text{true}}(\boldsymbol{X}) - h_{\text{TT}}(\boldsymbol{X}))^2\right] \leq \left(\frac{\sigma P}{p}\right)^2.$$

Furthermore, using the characteristic of expectation, if $\phi(x) = x^2$, which means $R_{\text{TT}}(h) = \mathbb{E}_{\boldsymbol{X},Y}[(F_Y(h(\boldsymbol{X})) - F_Y(Y))^2]$, we have

$$\begin{aligned} &R_{\text{TT}}(\hat{h}_{\text{TT}}) \\ &= \mathbb{E}_{\boldsymbol{X},Y}\left[(F_Y(\hat{h}_{\text{TT}}(\boldsymbol{X})) - F_Y(Y))^2\right] \\ &= \mathbb{E}_{\boldsymbol{X},Y}\left[(F_Y(\hat{h}_{\text{TT}}(\boldsymbol{X})) - F_Y(h_{\text{TT}}(\boldsymbol{X})))^2\right] + \mathbb{E}_{\boldsymbol{X},Y}\left[(F_Y(Y) - F_Y(h_{\text{TT}}(\boldsymbol{X})))^2\right] \\ &= \mathbb{E}_{\boldsymbol{X},Y}\left[(F_Y(\hat{h}_{\text{TT}}(\boldsymbol{X})) - F_Y(h_{\text{TT}}(\boldsymbol{X})))^2\right] + R_{\text{TT}}(h_{\text{TT}}). \end{aligned}$$

Since $(F_Y(y))' \geq p$, we have

$$\mathbb{E}_{\boldsymbol{X}}\left[(\hat{h}_{\mathrm{TT}}(\boldsymbol{X}) - h_{\mathrm{TT}}(\boldsymbol{X}))^2\right] \leq \frac{1}{p^2}\mathbb{E}_{\boldsymbol{X},Y}\left[(F_Y(\hat{h}_{\mathrm{TT}}(\boldsymbol{X})) - F_Y(h_{\mathrm{TT}}(\boldsymbol{X})))^2\right]$$

$$= \frac{1}{p^2}\left(R_{\mathrm{TT}}(\hat{h}_{\mathrm{TT}}) - R_{\mathrm{TT}}(h_{\mathrm{TT}})\right)$$

$$\leq O\left(\sqrt{\frac{\log 1/\delta}{n_{\mathrm{U}}}}\right) + O\left(\sqrt{\frac{\log 1/\delta}{n_{\mathrm{R}}}}\right)$$

with probability $1 - 3\delta$, where the last inequality holds from the same discussion as in Theorem 2. Note that $|\phi'(F_Y(h(\boldsymbol{x})))|, |F_Y(h(\boldsymbol{x}))\phi'(F_Y(h(\boldsymbol{x}))) - \phi(F_Y(h(\boldsymbol{x})))|$ are bounded since $F_Y(h(\boldsymbol{x})) \in [0,1]$ by definition. Combining these inequalities, we can see that

$$R(\hat{h}_{\mathrm{TT}}) \leq R(h_{\mathrm{true}}(\boldsymbol{x})) + 2\left(\frac{\sigma P}{p}\right)^2 + O\left(\sqrt{\frac{\log 1/\delta}{n_{\mathrm{U}}}}\right) + O\left(\sqrt{\frac{\log 1/\delta}{n_{\mathrm{R}}}}\right)$$

with probability $1 - 3\delta$. $\qquad\square$