[Reviews · NeurIPS 2019]

Reviewer 1



This paper proposes two novel approaches for the uncoupled regression problem, in which the correspondance between the input data and the targets is not known. Instead these methods use unlabeled data and pairwise comparisons of the targets. The first approach is the risk approximation approach (RA) and it minimizes an approximation of the risk defined based on the expected Bregman divergence. The second approach, called the target transformation (TT) approach, consists in mapping the target variable to a uniformly distributed random variable using the cumulative distribution function. Estimation error bounds are derived for each method. The empirical performances of the proposed methods are evaluated on synthetic data and on benchmark datasets. The paper is clearly written and well organized. To my knowledge, the proposed approach is novel. I think that the contributions of this paper are significant from a theoretical and also from an empirical point of view. Indeed the obtained results are convincing as they show that with many pairwise comparison data the proposed methods obtain close performances to supervised linear regression. I have a remark on the line 276: the authors claim that the TT approach outperforms the RA approach with sufficient pairwise comparison data however Figure 1 shows that the RA approach always obtains a smaller MSE than TT. Maybe there was some confusion with Figure 2 ? In addition the error bars on Figures 1 and 2 seem a bit stange. Minor comments: there is a typo on the line 255 (squared). Also I think that it should be the derivative of phi in the first term of Equation 7 and not phi. %%% Update: I have read the answers from the authors and I thank them for addressing some of my concerns, especially regarding the addition of other uncoupled regression methods in the comparison.

Reviewer 2



Authors propose a theoretically sound solution to uncoupled regression via pairwise ranking. They utilize statistical learning theory. They also analyze the estimators properties such as variance and develop generalization bounds. Furthermore they also supply empirical evidence for their algorithm. Paper is well written and connected to literature. Experimental study is well connected. My concerns are the practical value of the contribution and computational requirements,

Reviewer 3



The paper proposes two novel methods for uncoupled regression by employing pairwise comparison data within the framework of empirical risk minimization. Compared to existing methods, the new approaches can get rid of the strong assumptions on the target function by adopting the ideas of risk approximation or target transformation. The authors derive the estimation error bounds and demonstrate by numerical experiments that the proposed methods perform comparably to the ordinary supervised regression. The results are novel and very interesting. The paper is well written and easy to follow.

[Author Response · NeurIPS 2019]

We thank the reviewers for their thoughtful and useful feedback. Here, we address the main concerns raised by the reviewers. We will also fix minor typos in the final version of the paper.

**Adding comparison with other uncoupled regression:**    Since Reviewers 1 and 4 share the concern on the SVMRank baseline in the experiments, we answer it here. Strictly speaking, there is no uncoupled regression method that can be used in our setting to the best of our knowledge. As we stated in the paper, other uncoupled regression methods require external contextual data or strong assumptions that do not hold for the benchmark datasets that we used in our experiments. However, we may obtain some linear model by using the methods discussed in Hsu et al. [1] or Pananjady et al. [2], though their linear assumptions on the target function does not hold in our settings. We will add empirical comparison with these methods in the final version.

## To Reviewer 1

**The discussion about the comparison between the TT and RA approaches in Figure 1:**    Thank you for pointing it out. As you suspect, the discussion should have gone for Figure 2. We will fix it in the final version.

**The error bars seem strange:**    It is because we used log-plot in figures. Since the standard deviation of the TT method is large, the mean accuracy minus the standard deviation goes negative, which yields the strange shape in the log-plot. (Note that log of negative is treated as minus infinity.)

## To Reviewer 3

**Practical use of algorithm:**    To the best of our knowledge, there is no uncoupled regression method can be used in our setting. Hence, we compare our methods to the SVMRank benchmark, which is the closest to our setting. To make such a comparison meaningful, we decided to focus on the linear kernel. We would try different kernels in SVMRank or more complex models in the final version.

**Computational requirements:**    The loss function in the RA approach is just the sum of empirical means of the loss function, which does not take extra computation compared to ordinal empirical risk estimation methods. Moreover, by using the approximation described in Appendix A.1, the TT approach can be computed similarly to the logistic regression.

## References

[1] D. J. Hsu, K. Shi, and X. Sun. Linear regression without correspondence. In *Proceedings of the 30th Advances in Neural Information Processing Systems*, pages 1531–1540, 2017.

[2] A. Pananjady, M. J. Wainwright, and T. A. Courtade. Linear regression with shuffled data: Statistical and computational limits of permutation recovery. *IEEE Transactions on Information Theory*, 64(5):3286–3300, 2018.


[Meta-Review · NeurIPS 2019]

The reviews are all positive about the contributions of the paper and the reviewers also found the author feedback answered some of the issues raised in the initial reviews.